

# Towards unified reporting of genome sequencing results in clinical microbiology

Eugenio Mutschler[1,*], Tim Roloff[1,*], Aitana Neves[2],
Hege Vangstein Aamot[3], Belén Rodriguez-Sanchez[4], Mario Ramirez[5],
John Rossen[6], Natacha Couto[7], Ângela Novais[8,9],
Benjamin P. Howden[10], Sylvain Brisse[11], Sandra Reuter[12], Oliver Nolte[1],
Adrian Egli[1], Helena M. B. Seth-Smith[1] and the ESCMID Study Group
for Epidemiological Markers (ESGEM), and ESCMID Study Group for
Genomic and Molecular Diagnostics (ESGMD)

[1] Institute of Medical Microbiology, University of Zürich, Zurich, Switzerland
[2] Swiss Institute of Bioinformatics, Geneva, Switzerland
[3] Akershus University Hospital, Lorenskog, Norway
[4] Hospital Gregorio Marañon, Madrid, Spain
[5] Instituto de Microbiologia, Instituto de Medicina Molecular, Faculdade de Medicina,
Universidade de Lisboa, Lisbon, Portugal
[6] University Medical Center Groningen, Zwolle, Netherlands
[7] Centre for Genomic Pathogen Surveillance, Pandemic Sciences Institute, University of Oxford,
Oxford, United Kingdom
[8] UCIBIO. Applied Molecular Biosciences Unit, Department of Biological Sciences, Faculty of
Pharmacy, University of Porto, Porto, Portugal
[9] Associate Laboratory i4HB-Institute for Health and Bioeconomy, Faculty of Pharmacy,
University of Porto, Porto, Portugal
[10] University of Melbourne, Parkville, Australia
[11] Institut Pasteur, Paris, France
[12] Medical Center, University of Freiburg, Freiburg, Germany
* These authors contributed equally to this work.

Corresponding author
Helena M. B. Seth-Smith,
hsethsmith@imm.uzh.ch

## ABSTRACT

Whole genome sequencing (WGS) has become a vital tool in clinical microbiology, playing an important role in outbreak investigations, molecular surveillance, and identification of bacterial species, resistance mechanisms and virulence factors. However, the complexity of WGS data presents challenges in interpretation and reporting, requiring tailored strategies to enhance efficiency and impact. This study explores the diverse needs of key stakeholders in healthcare, including clinical management, laboratory work, public surveillance and epidemiology, infection prevention and control, and academic research, regarding WGS-based reporting of clinically relevant bacterial species. In order to determine preferences regarding WGS reports, human-centered design approach was employed, involving an online survey and a subsequent workshop with stakeholders. The survey gathered responses from 64 participants representing the above mentioned healthcare sectors across geographical regions. Key findings include the identification of barriers related to data accessibility, integration with patient records, and the complexity of interpreting WGS results. As the participants designed their ideal report using nine pre-defined sections of a typical WGS report, differences in needs regarding report structure and content across stakeholders became evident. The workshop discussions further highlighted the need to feature critical findings and quality metrics prominently in reports, as well as the demand for flexible report designs. Commonalities were

observed across stakeholder-specific reporting templates, such as the uniform ranking of certain report sections, but preferences regarding the depth of content within these sections varied. Using these findings, we suggest stakeholder-specific structures which should be considered when designing customized reporting templates. In conclusion, this study underscores the importance of tailoring WGS-based reports of clinically relevant bacteria to meet the distinct needs of diverse healthcare stakeholders. The evolving landscape of digital reporting increases the opportunities with respect to WGS reporting and its utility in managing infectious diseases and public health surveillance.

## INTRODUCTION

Whole genome sequencing (WGS) has emerged as a pivotal tool in clinical microbiology, extensively employed for molecular epidemiology in outbreak investigation and molecular surveillance, but also for diagnostic purposes to identify bacterial species, resistance mechanisms and virulence factors (*Meinel, Seth-Smith & Egli, 2017*). The complexity of WGS data necessitates specialized expertise for interpretation and reporting, catering to a diverse range of healthcare stakeholders. Variability among the stakeholders in WGS knowledge and understanding underscores the need for tailored post-analytical strategies to enhance reporting efficiency and generate an impact in order to justify the use of this, often costly, technology (*Tornheim et al., 2019*; *Crisan et al., 2018*).

Clinicians, often constrained by time, require WGS reports that concisely present essential information, including patient data, investigation results, and to a lesser extent methodological details (*Crisan et al., 2018*). Medical microbiologists are also likely to be interested in the technical details and quality assurance aspects within a report. Effective reports should integrate clear explanations and visual aids for rapid comprehension. Prior research recommends structuring reports to align with stakeholders' workflows, emphasizing key information, and avoiding abbreviations for clarity (*Tornheim et al., 2019*; *Crisan et al., 2018*; *Li et al., 2017*; *European Centre for Disease Prevention and Control, 2015*; *Cutting et al., 2016*; *Crisan, Gardy & Munzner, 2019*). The flexibility of report design is crucial to address varying needs across different healthcare sectors.

The medium of the report, whether printed or digital, significantly influences its content and design. Digital advancements, particularly in large language models (LLMs), offer in the near future transformative possibilities in report interaction and design, allowing for more detailed and interactive content that is currently difficult to incorporate in traditional printed formats (*Cutting et al., 2016*; *Nusrat, Harbig & Gehlenborg, 2019*; *Egli, 2023*). For instance, digital platforms can effectively display intricate and interactive figures, which are impractical for printed reports.

Previous studies on reporting of microbiological WGS results have predominantly focused on a universal template approach, mainly in the context of *Mycobacterium*

*tuberculosis* typing (*Tornheim et al., 2019*; *Crisan et al., 2018*). However, when considering the wider range of analyses from WGS, a dynamic reporting model, adaptable to specific stakeholders' backgrounds and needs, may more effectively facilitate lab-to-stakeholder communication, a crucial focus in the post-analytical part of diagnostics and surveillance. Tailoring reports to specific stakeholders, as seen in personal human genomics and patient-specific genetic disease confirmation, has been shown to significantly enhance understanding and engagement of both physicians and patients (*Shaer et al., 2015*; *Williams et al., 2018*). WGS reports for healthcare providers should include interpretative support and active clinical guidance for timely interventions, and at the same time facilitate effective communication with other healthcare workers or patients and their families (*Williams et al., 2016*).

This study aims to identify optimal WGS-based report structures for key stakeholder groups in clinical management, laboratory work, public surveillance and epidemiology, infection prevention and control, and academic research. Recognizing the current predominance of printed reports, we also explore stakeholders' preferences for information density in their respective reports. We focused on ISO accredited reporting in clinical microbiology, and the specific requirements therein (*International Organization for Standardization, 2022*). Collaborating with the ESCMID study groups on Genomics and Molecular Diagnostics (ESGMD) and Epidemiological Markers (ESGEM), our objectives are to design, standardize, and integrate these reporting recommendations, potentially for future implementation on online platforms such as the Swiss Pathogen Surveillance Platform (www.spsp.ch) (*Neves et al., 2023*; *Vears, Sénécal & Borry, 2017*).

## MATERIALS AND METHODS

*Study design*. This study aimed to develop optimized WGS-based reporting templates for various healthcare stakeholder groups, focusing on the structure and informational content. We adopted a human-centered design approach, engaging directly with stakeholders firstly through an online survey, followed by a workshop to discuss and adapt the recommendations.

*Participant selection*. Participants for the online-survey were recruited through snowball- and convenience-sampling *via* email using the ESGMD/ESGEM-network. At the end of the survey, interested participants were invited to share their contact details to join the following workshop.

*The survey*. Using the platform soscisurvey.de, we adapted our online survey from a previously published framework (*Crisan et al., 2018*). The survey is available in the Supplemental Materials. Targeted at five defined stakeholder groups-clinical management, laboratory work, public surveillance and epidemiology, infection prevention and control, and academic research - the survey consisted of five parts:

i) Professional background of participant, familiarity with reporting concepts, and data types (questions 1 to 4).
ii) Impact of reporting on clinical management, specifically for physicians/clinicians (questions 5 to 8).

iii) Surveillance focus, for epidemiologists, surveillance analysts, and researchers (questions 9 and 10).

iv) Designing an ideal report through a drag-and-drop interface featuring specific report sections (questions 11 to 14).

v) Optional: sharing contact information for future workshop participation (questions 15 to 19).

Participants could identify themselves with multiple stakeholder groups, with their choices directing them to relevant sections of the survey.

Part iv allowed participants to design their ideal report by ranking nine WGS-report sections, selected based on previous studies (*Tornheim et al., 2019*; *Crisan et al., 2018*; *Kozyreva et al., 2017*) (Sample Details, Short Summary of Final Results, Lineage/Organism, Drug Susceptibility/Resistance Prediction, WGS-based Cluster Detection, Assay Details, Sequence Depth and Coverage Details, Disclaimer regarding Limitations of the Analysis, Authorization of the Analysis/Signature) through a randomized drag-and-drop process to minimize bias. These sections are compatible with ISO accredited reporting in clinical microbiology (*International Organization for Standardization, 2022*). Participants could comment on their choices and indicate the desired level of detail ("block is not needed"/ "basic information is required"/"detailed information is required") for each section. The exact meaning of "basic" *vs.* "detailed" information was purposefully not defined. Instead, this difference and the content of the sections was aimed to be discussed during the following workshop and can be developed in future projects.

*Workshop.* The workshop's objective was to refine the report templates based on direct stakeholder feedback. The workshop was held remotely, and the key findings of the survey were discussed.

*Analysis.* We employed basic descriptive statistics for analyzing the survey data, presenting median values and interquartile ranges, except where noted otherwise. The survey's critical component was Part iv, where we focused on creating an ideal report structure and content tailored to different stakeholder groups. We based the report structures mostly on frequencies of ideal report structures. Essentially, the section that received the most votes for a particular position within the report would assume that position in the final report. When situations where multiple sections received an equal number of votes for the same position or where a single section was the top choice for multiple positions were encountered, we carefully examined which section received the most votes within each report section (beginning being ranks 1–3; middle ranks 4–6 and end ranks 7–9) and assigned its final rank accordingly. If any uncertainties remained, we discussed them within our working group and relied on expert opinions to determine the section's final ranking.

## RESULTS

*Overview of online survey and workshop.* Our mixed-methods study began with an online survey, receiving 64 valid responses (one duplicate response was excluded), followed by an

online workshop. This dual approach provided a comprehensive understanding of the needs of WGS reporting in microbiology and infectious diseases fields.

*Survey data analysis.* The survey's analytical focus of participants, their experience and needs, was within the responses to Parts i–iii. The survey attracted a diverse group of professionals, most of whom (59.4%) were affiliated with hospitals. The roles were as follows: clinical management ($n = 14$, 22%), laboratory work ($n = 44$, 69%), public surveillance and epidemiology ($n = 21$, 33%), infection prevention and control ($n = 14$, 22%), academic research ($n = 30$, 47%), and other ($n = 4$, 6%). 53% of participants identified themselves with more than one stakeholder group. The primary employers were: hospitals ($n = 38$, 60%), academic and research institutions ($n = 23$, 36%), reference laboratories ($n = 11$, 17%), public health organizations ($n = 10$, 16%), private clinics or primary care ($n = 0$, 0%), and other ($n = 5$, 8%). Despite diverse backgrounds of stakeholders, and some self-reported expertise in WGS related topics, there was a notable gap in expertise relating to biostatistics (27% lacking knowledge) and bioinformatics (22% lacking knowledge) (Table S1). Participants hailed from various regions, with the majority from Switzerland (35%), followed by the Netherlands (16%), Germany (10%), and Denmark (5%). Other European countries, the USA, Canada, Argentina, the Caribbean, and Qatar were also represented (Table S2).

## Current barriers in data reporting and utilisation

In clinical management, respondents indicated challenges in using WGS for diagnostics and treatment planning (Tables 1 and 2). The majority sought a faster turn-around-time and better integration with patient data. The complexity of interpreting WGS results was another concern.

Participants identified significant barriers in genomic data utilization for surveillance, including inconsistent data accessibility (63.2% citing technological costs as a major barrier) and difficulties in linking genomic data to patient records (Table 3).

## Ideal report structure

We asked participants to define the optimum report structure in Part iv of the survey, based around nine pre-defined blocks. This produced a clear general trend (Fig. 1), although the need for customization is emphasized as different stakeholder-groups gave some blocks different priorities. Within each stakeholder-group, participants expressed varied preferences (Fig. 2, Tables S3 to S7), with placement of blocks "Assay Details" and "Short Summary of Final Results" particularly variable. Additionally, the amount of detail required by each stakeholder-group within each block varied (Fig. 3). Further suggestions for data in the report included incorporating genetic location information for resistance genes (plasmid/chromosomal) and public health notifications.

*Workshop insights and integration.* The workshop was attended by 15 participants and conducted virtually, bringing together almost all stakeholder-groups for clarification and discussion. Unfortunately, representatives from the "Public Surveillance/Epidemiology" group were unable to attend, and the workshop was time-limited. However, one participant from the "Public Surveillance/Epidemiology" group sent written feedback to

**Table 1 Challenges when using WGS in diagnostics.**

| Challenges when using WGS in diagnostics | N (%) |
|---|---|
| I would like to receive data faster to make a more timely diagnosis | 10 (71.4%) |
| Important results come at different times and/or different documents | 6 (42.9%) |
| The lab data I receive is not routinely linked to patient data | 3 (21.4%) |
| I find it difficult to interpret the lab results I receive | 2 (14.3%) |
| No challenges – the lab data I currently receive is sufficient | 2 (14.3%) |
| I am not regularly receiving data that would help me to make a diagnosis | 2 (14.3%) |
| The lab data I currently receive does not help me to make a diagnosis | 0 |
| Other | 1 (7.1%) |

Note:
$N$ (%) are displayed for the 14 participants who belonged to the "Clinical Management" stakeholder group ($N = 14$). It was possible to choose several options.

**Table 2 Challenges when using WGS for treatment planning.**

| Challenges when using WGS for treatment planning | N (%) |
|---|---|
| Timeliness of results | 8 (57.1%) |
| Lab data is not routinely provided | 5 (35.7%) |
| Lab data is not linked to patient data | 2 (14.3%) |
| Results provided over multiple unconnected documents | 2 (14.3%) |
| Need for additional data | 2 (14.3%) |
| I do not encounter any issues. | 2 (14.3%) |
| Difficulty interpreting lab results | 1 (7.1%) |
| Other | 0 |

Note:
$N$ (%) are displayed for the 14 participants who belonged to the "Clinical Management" stakeholder group ($N = 14$). It was possible to choose several options.

**Table 3 Challenges to using genomic data more routinely in surveillance.**

| Challenges to use genomic data more routinely in surveillance | N (%) |
|---|---|
| Costs | 36 (63.2%) |
| Data not consistently accessible | 20 (35.1%) |
| It is not clear how to interpret this data for surveillance purposes | 10 (17.5%) |
| Data not consistently linked to patient data | 9 (15.8%) |
| It is not clear how this data is useful for surveillance | 5 (8.8%) |
| Difficulty interpreting lab results | 5 (8.8%) |
| Other | 17 (29.8%) |

Note:
$N$ (%) are displayed for the 57 participants which belonged to the stakeholder groups laboratory work, surveillance/epidemiology and infection prevention and control. It was possible to choose several options.

our survey which was discussed during the workshop. Key takeaways were: (i) emphasis on the clear representation of critically relevant findings ('red flags') and quality metrics at the beginning of reports for quick reference; (ii) the need for adaptable reports based on specific stakeholder requests, particularly in sections like "Sequence Depth and Coverage

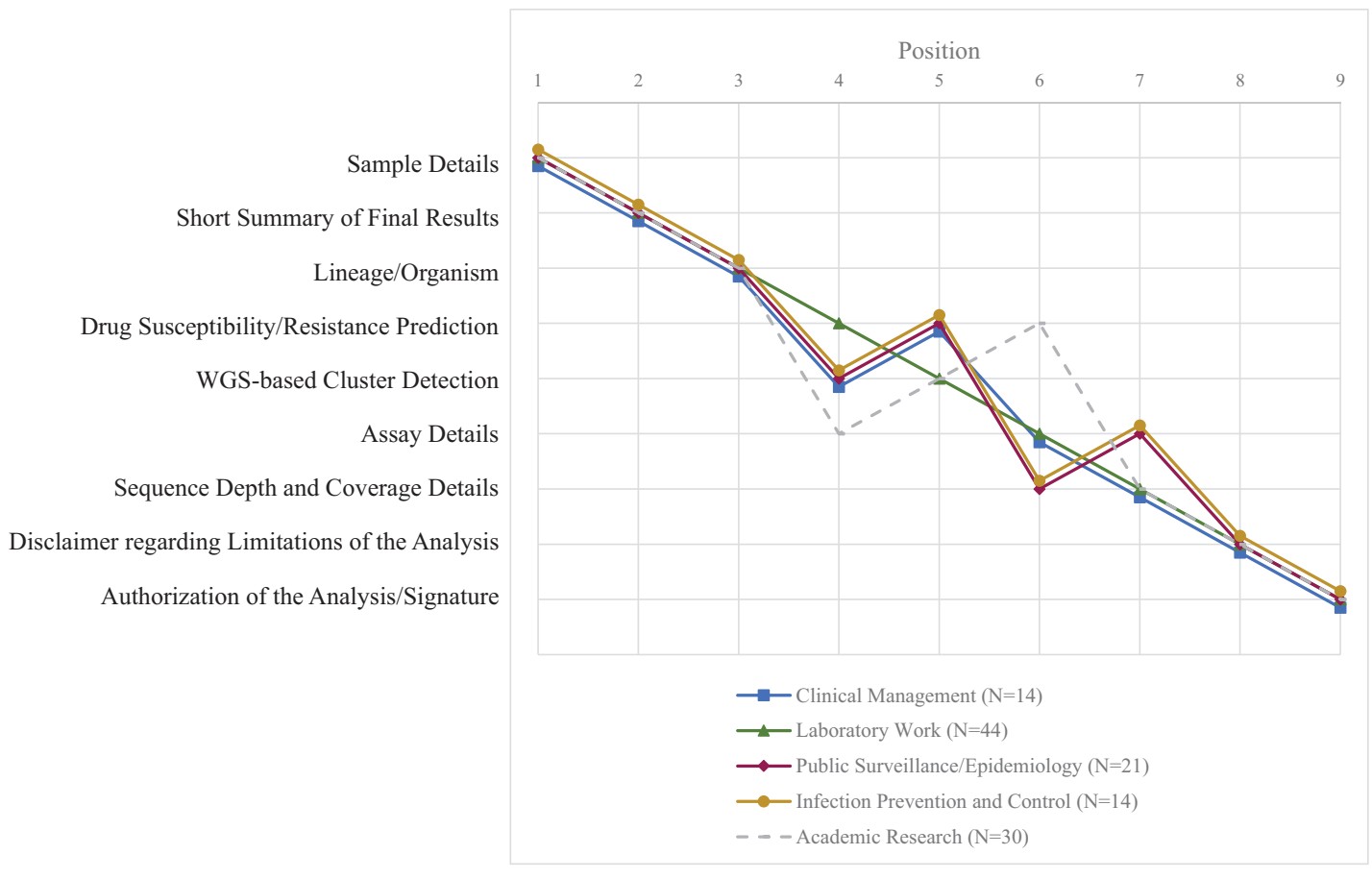

**Figure 1 Order of blocks.** Ideal order of nine suggested blocks inside a WGS-report, classified by stakeholder. "1" corresponds to the beginning of the report, "9" to the end. Position by stakeholders (defined in the legend): modal opinions are represented (see Methods).

Details" and "Drug Susceptibility"; and (iii) integration of the report data into an electronic patient health record (EPHR).

## DISCUSSION

WGS has established itself as a crucial tool in outbreak investigation and molecular surveillance for public health, but also increasingly for diagnostics. The COVID-19 pandemic in particular underscored its importance (*Neves et al., 2023*). Effective WGS implementation relies heavily on structured and insightful reporting, which should ideally be tailored to the different stakeholders in healthcare. The findings also underscore the necessity for faster, more integrated, and user-friendly data reporting systems in both clinical and research settings (*López Yeste et al., 2021*). These insights are pivotal in developing effective WGS reporting tools, enhancing the utility of genomic data in managing infectious diseases and public health surveillance.

Our study has highlighted the diverse structuring and informational requirements across different stakeholder groups, revealing both common and unique preferences within reporting templates queried in this study. For the first time, concrete suggestions are

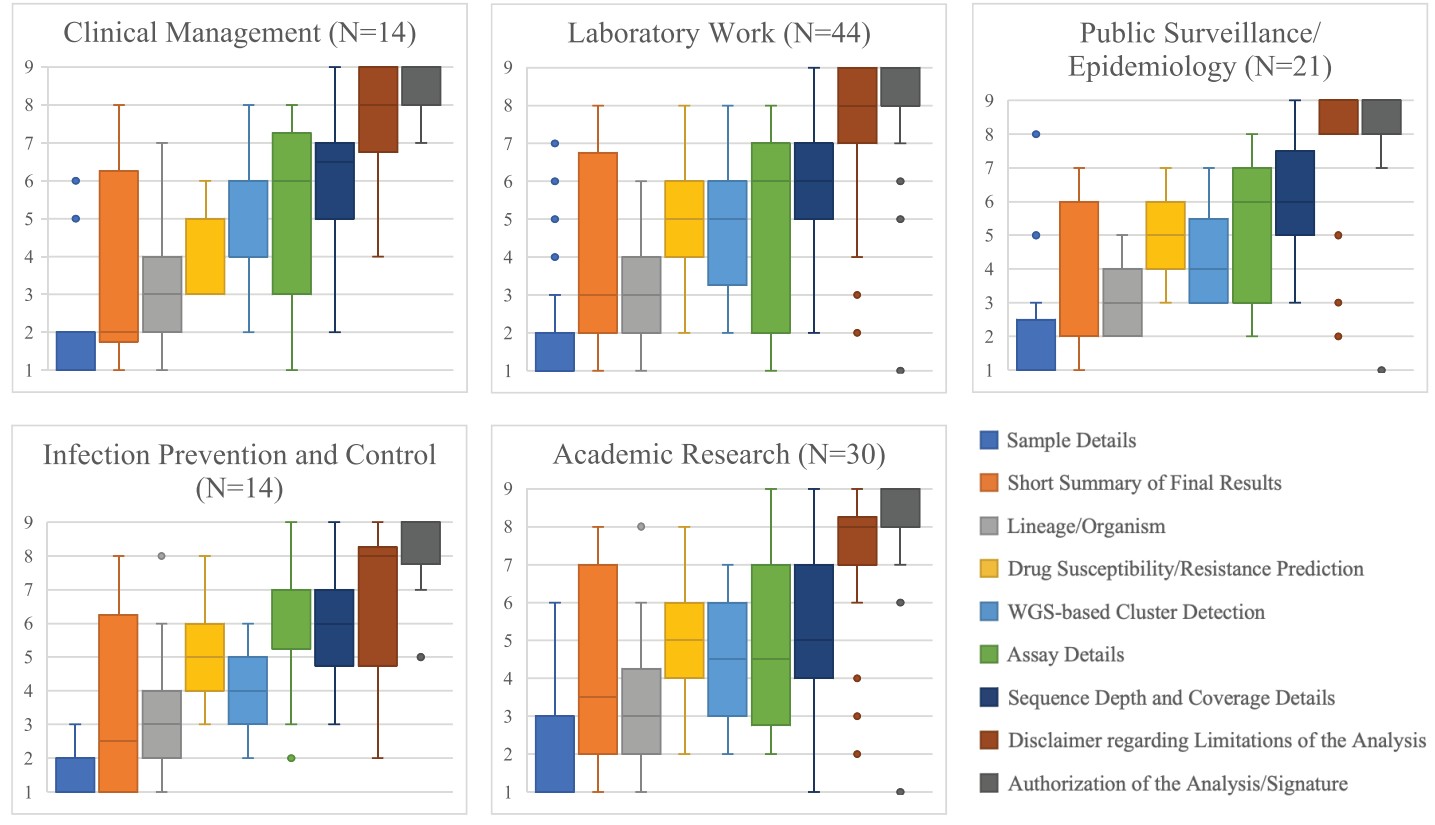

**Figure 2 Variation of participant opinions within stakeholder groups regarding the order of blocks.** The chart elements convey specific information: boxes for the interquartile ranges, bars symbolize the median value, whiskers depict the minimum and maximum values, and points highlight any outliers.                                       

made on how to individualize the reporting structure and content depending on different key recipients of the healthcare system. Each stakeholder group, from clinical management to academic research, displayed somewhat distinct preferences in content, reflecting their unique operational and informational needs. In our definition of five stakeholder-specific reporting structures, some commonalities such as the uniform ranking of certain sections (Sample Details, Short Summary of Final Results, Lineage/Organism, *etc.*) were identified. However, preferences of the order for report sections "Drug Susceptibility/Resistance Prediction" and "WGS-based Cluster Detection" varied, with notable exceptions in the "academic research" group. Awareness of these preferences, and development of relevant templates may also decrease reporting turn-around-time, and improve ease of integration with data management systems.

The order of content within a specific report may also depend on the initial reason for submitting samples for WGS: evaluation of a treatment option or an outbreak investigation. Although not specifically addressed in our study, this could be extended to reporting species-specific results in order to best represent the huge biological diversity of pathogens. An example of reporting template system tailored for *Klebsiella* and adaptable to different end-users is available within the BIGSdb platform (https://bigsdb.readthedocs.io/en/latest/data_analysis/reports.html#reports). Also noteworthy is the demand to have

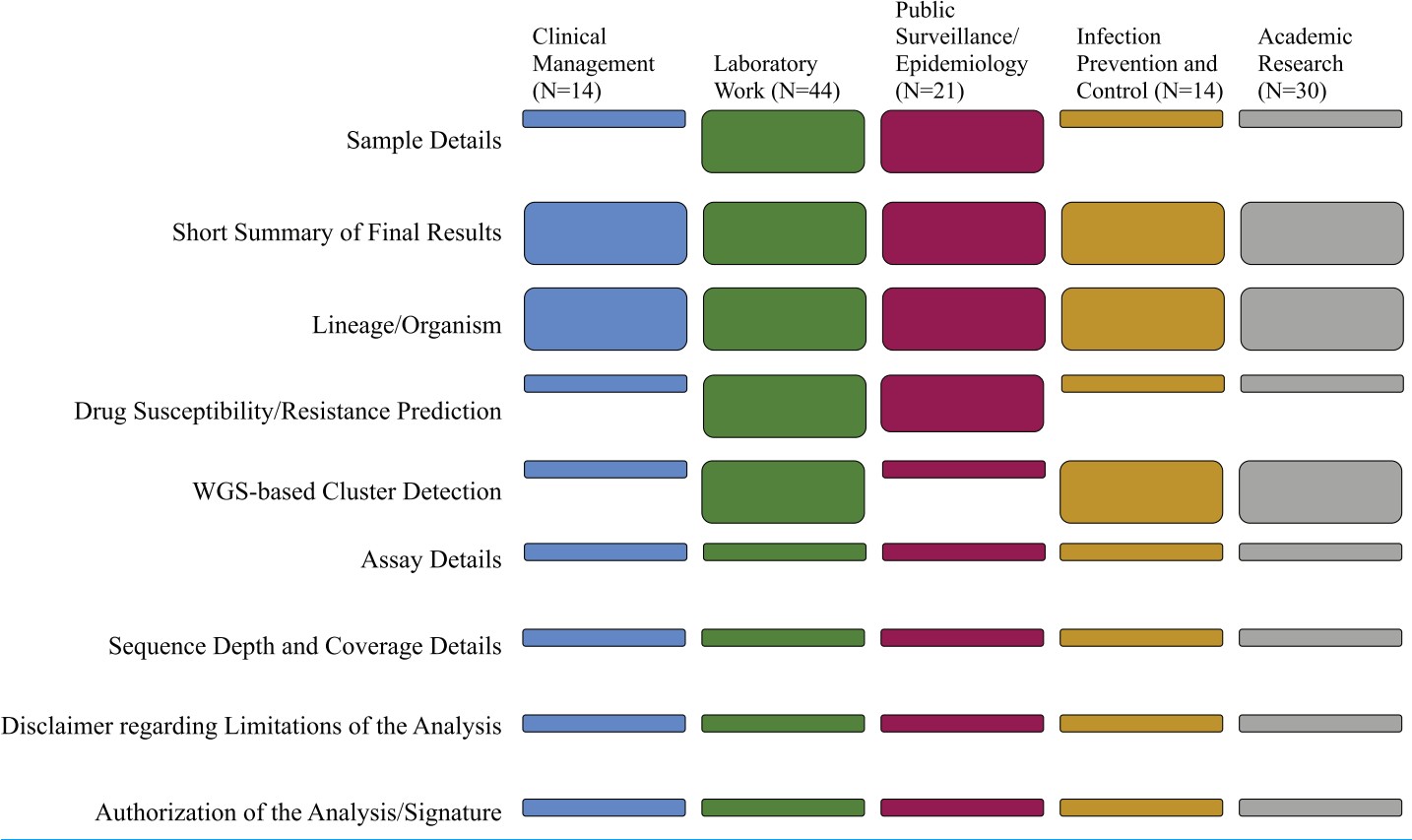

**Figure 3 Desired amount of information inside each block depending on the stakeholder.** A thinner block indicates "basic information", a thicker block indicates "detailed information needed".

more detailed technical background information for stakeholders associated with "Laboratory Work". Indeed, the requirements for more or less detailed blocks among different stakeholders was interesting, and deserves further exploration.

Despite the variations, the similarity across templates likely reflects a shared goal in managing infections and outbreaks. The familiarity of these formats within participant organizations suggests a preference for established reporting practices. Nonetheless, even minor adjustments in section ordering can significantly enhance the accessibility and relevance of information. As such, report structure may be best optimised to reflect specific requests as well as stakeholder origins.

The need for dynamic report designs that cater to specific requests and stakeholder needs was a recurring theme. These insights are pivotal in developing effective WGS reporting tools, enhancing the utility of genomic data in managing infectious diseases and public health surveillance.

With the advent of digital reporting solutions, the landscape of WGS reporting is set to evolve significantly (*Egli, 2023*). Modern LLMs are poised to revolutionize post-analytic interpretation, offering more dynamic, interactive, and adaptable reporting formats (*Egli, 2023*). This advancement necessitates research into the integration of reports with

electronic health records and the development of formats that cater to both digital and traditional methods.

*Limitations.* This study, while providing valuable insights into the structuring and content preferences for WGS-based reporting across various healthcare stakeholders, has several limitations that warrant consideration. First, the majority of our survey participants self-reported as being somewhat skilled in WGS-related fields, which is not the case for all relevant report recipients, potentially skewing the results towards more technically proficient viewpoints. Secondly, the absence of representatives from primary healthcare and private practices at the survey and the workshop, and public health authorities at the workshop, limits the generalizability of our findings to these crucial sectors of healthcare. Future studies would benefit from a more diverse participant pool, including stakeholders with varying levels of expertise in WGS. Thirdly, many participants identified with multiple stakeholder groups, complicating the interpretation of results. This overlap might have influenced the preferences expressed for report content and structure, as individuals may bring biases from their dominant professional roles. Disentangling these influences remains a challenge and should be addressed. Finally, our approach employed descriptive statistics, which, while effective for initial insights, lacks the depth that advanced statistical methods could provide. More sophisticated analyses on a larger dataset could uncover nuanced trends and correlations that our study may have missed.

The study considered printed report formats, overlooking the rapidly evolving domain of digital reporting solutions. As digital platforms offer greater flexibility and interactivity, we need to emphasize the development and evaluation of digital WGS reporting, particularly in the context of advancements in LLMs. Of note, there are no FDA or IVDR-approved LLMs for WGS-based reporting available yet. The fast-paced advancements in digital reporting technologies and LLMs present a moving target for research (*Egli, 2023*).

## CONCLUSIONS

Our findings indicate important but subtle variations in needs for WGS-based reporting among different stakeholder groups, concerning both the structure and the depth of information within defined report sections. Specifying the content for each block is a key step, but will also depend on individual requests, and the requirements of accreditation. The emergence of digital solutions introduces new possibilities and challenges in presenting and securely transmitting WGS results, calling for research into larger and more diverse stakeholder populations, including those with less WGS knowledge. Adapting WGS reports according to the recipient's familiarity with WGS-related topics is important to enhance their understanding of the results. Therefore, we recommend further research into the specific needs of individuals with different levels of expertise in WGS-related fields.

This project represents an initial step in understanding the requirements and preferences of WGS report recipients. The community driven efforts with support of the

ESGMD and ESGEM study groups was crucial to access a broad range of opinion leaders in the field. With continued research and development, including harnessing the capabilities of LLMs for digital reporting, we can enhance the comprehension and utility of WGS results in clinical microbiology.

## ACKNOWLEDGEMENTS

We thank all the participants who contributed with their valuable comments to the initial survey and also during the workshop with the in-depth discussion to further shape the standardization of WGS-based reporting. "The ESCMID Study Group for Epidemiological Markers (ESGEM), Basel, Switzerland: Hege Vangstein Aamot, Mario Ramirez, John Rossen, Natacha Couto, Ângela Novais, Benjamin P. Howden, Sylvain Brisse, Sandra Reuter, Adrian Egli, Helena MB. Seth-Smith. The ESCMID Study Group for Genomic and Molecular Diagnostics (ESGMD), Basel, Switzerland: Hege Vangstein Aamot, Belén Rodriguez-Sanchez, Mario Ramirez, John Rossen, Natacha Couto, Ângela Novais, Adrian Egli".

### Funding

This study was funded by an unrestricted research grant to Adrian Egli by the University of Zurich, Switzerland. The funders had no role in study design, data collection and analysis, decision to publish, or preparation of the manuscript.

### Grant Disclosures

The following grant information was disclosed by the authors:
Unrestricted research grant to Adrian Egli by the University of Zurich.

### Competing Interests

The authors declare that they have no competing interests.

### Author Contributions

- Eugenio Mutschler conceived and designed the experiments, performed the experiments, analyzed the data, prepared figures and/or tables, authored or reviewed drafts of the article, and approved the final draft.
- Tim Roloff conceived and designed the experiments, performed the experiments, analyzed the data, authored or reviewed drafts of the article, investigation, and approved the final draft.
- Aitana Neves performed the experiments, authored or reviewed drafts of the article, investigation, and approved the final draft.
- Hege Vangstein Aamot performed the experiments, authored or reviewed drafts of the article, investigation, and approved the final draft.
- Belén Rodriguez-Sanchez performed the experiments, authored or reviewed drafts of the article, investigation, and approved the final draft.

- Mario Ramirez performed the experiments, authored or reviewed drafts of the article, investigation, and approved the final draft.
- John Rossen performed the experiments, authored or reviewed drafts of the article, investigation, and approved the final draft.
- Natacha Couto performed the experiments, authored or reviewed drafts of the article, investigation, and approved the final draft.
- Ângela Novais performed the experiments, authored or reviewed drafts of the article, investigation, and approved the final draft.
- Benjamin P. Howden performed the experiments, authored or reviewed drafts of the article, investigation, and approved the final draft.
- Sylvain Brisse performed the experiments, authored or reviewed drafts of the article, investigation, and approved the final draft.
- Sandra Reuter performed the experiments, authored or reviewed drafts of the article, investigation, and approved the final draft.
- Oliver Nolte performed the experiments, authored or reviewed drafts of the article, investigation, and approved the final draft.
- Adrian Egli conceived and designed the experiments, performed the experiments, authored or reviewed drafts of the article, investigation, and approved the final draft.
- Helena M. B. Seth-Smith conceived and designed the experiments, performed the experiments, prepared figures and/or tables, authored or reviewed drafts of the article, investigation, and approved the final draft.

### Data Availability

Raw data are available in the Supplemental Files.

### Supplemental Information

Supplemental information for this article can be found online at http://dx.doi.org/10.7717/peerj.17673#supplemental-information.

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
