# Peer review of "Towards unified reporting of genome sequencing results in clinical microbiology"

_PeerJ, doi:10.7717/peerj.17673_

## Round 0.1 · original submission · Major Revisions

The authors are requested to carefully revise the manuscript and answer the questions raised by the reviewers.

Reviewer 1 ·

Basic reporting

no comment

Experimental design

no comment

Validity of the findings

no comment

Additional comments

The manuscript investigated the needs of whole-genome sequencing (WGS) reporting in microbiology and infectious diseases from key stakeholders through an online survey and a workshop. By better understanding of the reporting needs, the aim is to reduce the complexity of WGS data interpretation by stakeholders and facilitate the decision making. I found this manuscript is pretty interesting, and may attract attention of not only people designing WGS reports but also people performing WGS data analysis. Therefore, I recommend the publication of this manuscript.
Comments:
1. It would make the manuscript more readable by including a summary table highlighting key survey/workshop findings.
2. Could the authors provide specific examples or cases to illustrate how the findings in this manuscript can help improvement of existing WGS reports?

Reviewer 2 ·

Basic reporting

No comment.

Experimental design

The methods are explained with sufficient details, though some points still need clarification:

Line 151: How is the “region” (top/middle/bottom) defined and how the finalized ranking is calculated? This aspect seems not to be mentioned in the survey design.

The survey:
1. Q5, 6 and 10 have several options that are essentially the same, but all worded differently. Also, the descriptions of Q5 and Q6 don't follow the same format. According to Table 1 & 2, these are intended to be very clear-cut questions on diagnosis and treatment planning, respectively. This may introduce unwanted bias or cause the questions to overlap (as the participant might misunderstand the difference between questions, or consider options that are essentially the same as different). Could the authors explain why the wordings are different?

2. The questions on challenges in diagnosis/treatment/surveillance are an important part of the survey, but both the questions and the options seems not to reflect the needs regarding the design of reports, which is the main focus of the study. For example, challenges like timeliness, accessibility and linkage to patient fall within the responsibility of the data itself, not the reporting, and the sections in part iv could not address these challenges either. Could the authors elaborate more on this part of the survey design, and how the findings may help improve the reporting?

Validity of the findings

Overall, the results are comprehensive and well-presented. However, I feel some more additional details and discussions are needed to gain more insights. For example:

Lines 135~139: As stated here, the definition of "basic" and "detailed" is a bit ambiguous and also varies across sections. It should be discussed and clarified in the workshop discussion, but that discussion is lacking in the results part.

Line 174~181: First of all I feel the discussions of Table 1~3 should belong to the same section. Also, as mentioned in previous comments, it would be better to further explain how these findings could directly guide the design of reports.

Lines 220~224: Could the authors elaborate on how the purpose of the WGS sample submission may affect the order of the report (e.g. which purposes may lead to which order preferences), and how the conclusion could be extended to species-specific results?

Table 3: There are quite a few "other" responses. By looking at the raw survey results, some responses are similar to the options, and some could belong to a new category of "standardization/comparison of the pipelines/methods" (or other more accurate wordings) which is not present in the options. Could the authors also discuss a bit more about these results?

Additional comments

No comment.

---

## Round 0.2 · accepted · Accept

After revisions, all reviewers agreed to publish the manuscript. I also reviewed the manuscript and found no obvious risks to publication. Therefore, I also approved the publication of this manuscrip

t.

Reviewer 1 ·

Basic reporting

no comment

Experimental design

no comment

Validity of the findings

no comment

Additional comments

I have no further comments on the revised manuscript.

Reviewer 2 ·

Basic reporting

No comment

Experimental design

No comment

Validity of the findings

No comment

Additional comments

No comment